# Measuring the success of programmes of care for people living with dementia: a protocol for consensus building with consumers to develop a set of Core Outcome Measures for Improving Care (COM-IC)

Tracy Comans,[1] Kim Nguyen [ID],[1,2] Len Gray,[1] Leon Flicker,[3] Paula Williamson,[4] Susanna Dodd,[4] Anna Kearney,[4] Colm Cunningham,[5,6] Thomas Morris,[5,7] Jack Nunn,[8,9] Dominic Trepel,[2] Osvaldo P Almeida [ID],[3] Danelle Kenny [ID],[1,10] Alyssa Welch,[1] Judy A Lowthian [ID],[11,12] John Quinn,[1,13] Glenys Petrie,[1,13] Tiet-Hanh Dao-Tran,[1] Asmita Manchha,[14] Susan E Kurrle[5]

For numbered affiliations see end of article.

**Correspondence to**
Dr Tracy Comans;
t.comans@uq.edu.au

## ABSTRACT

**Introduction** The Core Outcome Measures for Improving Care (COM-IC) project aims to deliver practical recommendations on the selection and implementation of a suite of core outcomes to measure the effectiveness of interventions for dementia care.

**Methods and analysis** COM-IC embeds a participatory action approach to using the Alignment–Harmonisation–Results framework for measuring dementia care in Australia. Using this framework, suitable core outcome measures will be identified, analysed, implemented and audited. The methods for analysing each stage will be codesigned with stakeholders, through the conduit of a Stakeholder Reference Group including people living with dementia, formal and informal carers, aged care industry representatives, researchers, clinicians and policy actors. The codesigned evaluation methods consider two key factors: feasibility and acceptability. These considerations will be tested during a 6-month feasibility study embedded in aged care industry partner organisations.

**Ethics and dissemination** COM-IC has received ethical approval from The University of Queensland (HREC 2021/HE001932). Results will be disseminated through networks established over the project, and in accordance with both the publication schedule and requests from the Stakeholder Reference Group. Full access to publications and reports will be made available through UQ eSpace (https://espace.library.uq.edu.au/), an open access repository hosted by The University of Queensland.

## INTRODUCTION
### Background and rationale
Dementia is a syndrome characterised by cognitive and functional decline, posing unique and complex challenges for health systems globally.[1] Worldwide, there are over

## STRENGTHS AND LIMITATIONS OF THIS STUDY

⇒ A key strength of this study is the use of participatory codesign and coevaluation methods to embed people with an experience of dementia in the selection of core outcome measures for interventions that affect them.
⇒ This study applies the Alignment–Harmonisation–Results framework to develop and assess feasibility of a set of core outcome measures for dementia care.
⇒ Feasibility will be tested through implementation within partner organisations.
⇒ Limitations are imposed by the short timeframe for the feasibility study.

55 million people living with dementia, with significant diversity in symptoms, types and demographics. Experiences vary widely across many features of the disease, including age of onset, symptoms, severity, duration and progression. This variability and associated uncertainty demand equal variability in support services and care delivery models, spanning across all care settings and multiple providers. Under these conditions, evaluation models vary widely,[2 3] and it is increasingly difficult to compare interventions and ascertain the value and impact on quality of care of one intervention in relation to another.

The delivery of interventions and care models across different settings and for different populations, combined with the capability and capacity of service organisations, naturally leads to the use of a wide

range of instruments and approaches to report outcomes and to measure success (clinical efficacy and effectiveness, care quality and efficiency), despite their shared focus on dementia. The use of different assessment instruments reduces the comparability of results across care models, leading to the slow translation of evidence into practice and reduced capacity to improve quality of care. It also results in fragmentation and duplication of service provision across providers, creating system-wide inefficiencies. In Australia, there is no recommendation or mandatory reporting of outcomes relating specifically to the provision or quality of care for people living with dementia, despite the increasing burden of disease associated with dementia in older people (85+ years).

There is a consensus among stakeholder groups that inefficiencies in dementia care should be minimised, and that meaningful outcome data that are consistent, comparable and feasible are crucial to improving quality of care.[4 5] Core outcome sets (COS) are designed to provide consistency and comparability, though they are not yet implemented and evaluated at a policy level or as a form of national benchmarking, and the feasibility of this approach is not currently known. This research aims to bridge this gap by evaluating existing core outcomes used for monitoring quality of routine dementia care around the world, and adapting these to fit routine data collection within the Australian care setting.

Changing long-term data collection in industry (ie, outside the controlled trial environment) poses challenges relating to feasibility and consensus. It is not always possible or economically viable to change the type or amount of routinely collected data. To maximise the feasibility of collecting and reporting COS, this research will engage with industry partners to identify and evaluate the measures currently routinely collected in the Australian aged care setting. The challenge of reaching consensus will be addressed by engaging multiple stakeholders with varying perspectives on the importance of different outcome measures using a codesigned research design guided by participatory action research (PAR) principles.

Involving stakeholders in the design and development of this research will enhance the quality of research outcomes by increasing the relevance of selected measures to people affected by dementia.[6] Statements from WHO proclaim that people with lived experience of dementia have the greatest direct interest in research outputs and are therefore central to the research process and should be involved in all stages.[1] Involving people in research means conducting research with people, as opposed to 'on' or 'for' people.[7] People with cognitive impairment are capable of contributing equally to decisions about their care and have the right to voice their preferences extending beyond individual care planning into policy and research priorities.[8] The involvement of people affected by dementia in the identification of core outcomes to measure the quality of their care creates a balance between individual autonomy and collective community values that facilitates equitable access to health and care services.[6 9 10]

The objective of this research is to develop, test and provide implementation recommendations for a COS applicable to routine care for people living with dementia. The COS will be derived from an international literature search, current industry routine data collection and outcomes considered important by a diverse range of stakeholders. Consensus will be sought through a variety of PAR activities including meetings, discussion boards and consensus surveys.

The COS developed during this research will describe a minimum level of routinely collected information that is critical to the determination of quality of care provided to people living with a diagnosis of dementia. The COS will be trialled in aged care settings that already routinely collect data and regularly deliver mandatory reports to government. Establishing feasibility here provides the potential to disseminate the COS to additional care settings with an interest in improving quality of care for people living with a diagnosis of dementia.

At a national level, this research—the Core Outcome Measures for Improving Care (COM-IC) project—is aligned with several recommendations made in 2021 by the Royal Commission into Aged Care Quality and Safety. Specifically, the project aligns with: requirements for ongoing research into the development and use of an evidence base for quality indicators; publication and guidance on data use; methodology for benchmarking; setting improvement targets; and public reporting of performance against measurable data.[11]

The aim of this protocol is to outline the research plan for COM-IC. This encompasses the development of a process for involving people living with dementia in decisions about the measurement of dementia care and using participatory methods to facilitate working partnerships with stakeholders, enabling timely translation of evidence-based research into practice.

## METHODS AND ANALYSIS
### Study setting
The majority of people living with dementia seeking care will access services in the aged care setting, either in residential care or in community-based aged care programmes.[8 12] These organisations routinely collect and report outcome measures, meaning they are well positioned to accept and implement recommendations for core outcome measurement. As such, this research is positioned in the aged care setting, incorporating both residential aged care and home care services.

### Methods
The research design is a multicomponent study that applies the Alignment–Harmonisation–Results (AHR) framework[13] combined with Participatory Action Research (PAR) methodology prioritising stakeholder engagement and co-design[14] to develop and implement a

core set of outcome measures relevant to the provision of high-quality routine care to people living with dementia. Specific methods for sourcing, evaluating and ultimately recommending outcomes are unknown at the protocol stage, as they will be codesigned as a collaboration between the international investigators and the Stakeholder Reference Group (SRG). It is anticipated that relevant outcomes will be identified through an international scoping review of existing core outcome sets and presented to stakeholders using the same taxonomy as the scoping review to develop understanding of outcomes and outcome measurement. This structure will then be adapted and expanded based on additional information such as outcomes collected by registries, by industry in current practice, and outcomes that are considered important by stakeholders.

### Participatory action research

PAR is an approach to research that emphasises the engagement and empowerment of research participants, sharing the responsibility for decision-making with those most impacted by the research at each stage of the research process, including design, methods, data collection, analysis and reporting.[15] From inception, the COM-IC project has involved people with lived experience of dementia, as consultation with this community is fundamentally critical to generating research that meets the needs of healthcare recipients. By design, this approach limits the ability to plan and define specific methodologies for the life of the project, as the design requires stakeholder input. Consequently, specific and granular details are not able to be described at the protocol stage, instead they develop organically through collaborative processes and are incorporated into the research protocol and ethics approval via amendments.

### AHR framework

The AHR framework is a model developed to improve resource allocation in global aid, a system with similar complexities to healthcare quality.[13] The COM-IC project

adapts this framework to fit the dementia care context (figure 1).

### Alignment

Alignment is achieved through reaching agreement on a suite of standardised (core) outcomes that define a baseline or minimum standard for all dementia-related interventions and care delivery models in Australia. As there is no consistent approach to quality and outcome measurement for dementia care across all settings, reaching agreement first involves stakeholders and international research partners collating existing quality measures for people living with dementia, including those routinely collected by our industry partners. This list will be compiled from various sources across research, industry and stakeholder groups.

Information on existing data collection will be included alongside the results of a scoping review of international COS for dementia, using the Core Outcome Measures in Effectiveness Trials database, including their application and use,[3 16 17] commonly applied to trials rather than routine care. The SRG will review the complete list of measures and develop a consensus on stakeholder preferences.[18] A gap analysis will converge research, industry and stakeholder viewpoints, providing clarity around outcome measures that are relevant to the provision of high-quality dementia care, and are feasible to implement in the Australian context.[2]

Identified outcome measures with high relevance to people living with dementia will be assessed for methodological quality and relevance to the population and setting (table 1).

Subsequently, this information will be summarised in plain English as a decision-making tool for the SRG. A modified e-Delphi method will be applied to reach consensus, using the STARDIT (Standardised Data on Initiatives) tool as the primary reporting mechanism.[18–20] The definition of consensus will be confirmed with the SRG using discussion threads on the Loomio

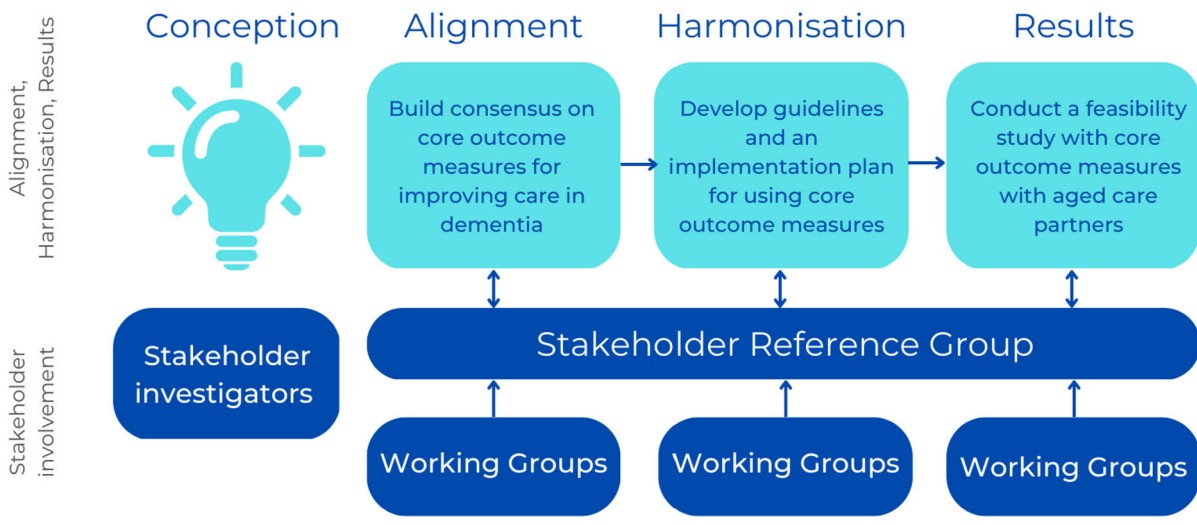

**Figure 1** Alignment–Harmonisation–Results (AHR) framework.

**Table 1** Criteria for inclusion of outcome measures

| Criterion | Description |
| --- | --- |
| Clarity | An outcome measure should be specific, stating clearly what is being measured, how it is measured and its interpretation. |
| Validity | The outcome measure should measure what it intends to measure (ie, it discriminates between good and bad performance), is not subject to large variation due to random changes in small numbers of events and minimises potential bias. |
| Measurability and timeliness | The performance aspect should be measurable and quantifiable. |
| Consistency over time | The outcome measure should measure quality aspects consistently over time and in a timely manner. |
| Added value | The outcome measure should capture elements of performance not captured elsewhere. |
| Measure of value (benefit) | The outcome measure should capture aspects of performance proven to be valued by users and providers of services. |
| Attribution and granularity | The outcome measure should measure aspects of performance that can be attributed to healthcare or aged care settings, and that are subject to healthcare or aged care system control. |
| Ease of collection | The outcome measure must be able to be collected within usual practice in settings where it is required to be collected or minimise additional burden on collectors if this is not possible. |

Modified from the Accounting for the Quality of NHS Output framework.[35]

platform. Deidentified, aggregated results from survey tools combining Likert scales and ranking exercises will be reported back to the group, with provision for discussions, questions and answers through a combination of synchronous and asynchronous processes. Rounds will continue until the agreed definition of consensus is reached. Outcomes identified as core will be transferred to the recommendations, codesigned with stakeholders. The STARDIT approach uses structured communication techniques applied systematically to stakeholder panels through several rounds (internet-based surveys) until consensus is reached.[20] Each round is summarised by the group facilitator and results reported to all participants before the next round. Rounds continue until consensus is achieved and the measures with consensus will become the core outcome measures recommended in the recommendations. It is anticipated that the group will adopt a divergent-convergent approach to building a list of outcomes. Each list from identified sources will be considered for inclusion and evaluated against the NHS Quality Output Framework.

This approach is designed to minimise the likelihood of missing data during the consensus-building process. If missing data are identified, the primary course of action will be to afford the participant further opportunity to offer their opinion.

## Harmonisation
Harmonisation refers to the generation of a standardised set of recommendations for core outcome measures in dementia care. Following the GRADE (Grading of Recommendations Assessment, Development and Evaluation) method, these recommendations will include methods to appraise, monitor and evaluate interventions and models of care (eg, functional decline, economic evaluation, societal benefit cost analyses), or methods to measure (key) performance indicators at the organisation and system levels (eg, composite indicators for quality of care and operational efficiency at the organisation level, or equality/disparity of access and utilisation at the sector level).[21] They will also include guidance for the selection of core outcomes for well-defined target populations (eg, people with younger onset dementia), types of interventions, care settings and sectors. The recommendations will be accompanied by a recommended implementation plan that addresses potential barriers to implementation identified through the Alignment phase, a clear timeline and stakeholder responsibilities.

A Recommendation Development Group will be established that comprises members of the research team, SRG and representatives from the Australian Department of Health and Aged Care.

The Harmonisation phase will include preparation for the feasibility study (to be conducted in the Results phase). Outcomes of the gap analysis will be used to identify data that are not currently collected by partner organisations. The feasibility of data collection will be determined through focus groups and interviews conducted internally with partner organisations' staff, care recipients and family carers to understand whether additional data collection is acceptable, and the most practical methods of collection. An adoption timeline will be developed, and a subset of outcome measures identified that are deemed feasible for trial implementation within the partner organisation.

## Results
The Results phase will comprise a feasibility study with industry partners that focuses on the translation of research into practice. Data relevant to core outcome measures will be extracted from partner organisations' databases before and after study and evaluated to determine both the functionality and usability of the outcome measures. A qualitative evaluation of the implementation experience will determine the practicality and

sustainability of the recommendations and implementation plan developed in the Harmonisation phase.

Representatives from two large Australian aged care providers are embedded in the COM-IC investigation team. They will support the overall project by providing information on current data collection during the Alignment phase, and support their organisations to implement the core outcome recommendations in the Results phase. Each partner organisation will trial the implementation of recommended measures that can be collected within existing data collection systems for 6 months. This may include modification of existing instruments to collect additional information relating to new recommended indicators.

The recommendations and implementation plan will be coevaluated by the SRG and investigation team, and revised based on outcomes of the feasibility study.[22] Any identified corrections will be made to produce a final set of recommendations.

### Patient and public involvement (stakeholder engagement)

Methods used to involve people affected by dementia in this project have been informed by the PAR paradigm.[23] This approach was selected following consultation with people with lived experience of dementia. The PAR paradigm guides stakeholder involvement alongside the AHR framework by embedding participation and action in the research process. Involvement refers to using participatory methods such as workshops, focus groups, Delphi panels, surveys, policy exercises, and mutual learning activities to facilitate inclusion of stakeholders with previously limited involvement in research activities like development, design, conduct, analysis, interpretation, dissemination, implementation and evaluation.[14 24]

### Dynamic learning needs assessment

The premise of PAR and codesign is that stakeholders are experts of their lived experience, while researchers are experts in a particular field of study, and each brings specific knowledge to the research process that produces a more nuanced response to a research question. As the application of PAR methodology for people living with dementia is relatively recent and rare, particularly for the length of time stakeholders are asked to commit to COM-IC, we intend to evaluate the effectiveness of PAR methods, and investigate changes that participation has on attitudes, beliefs and understanding of codesign research through a dynamic learning needs assessment. The assessment is a codesigned survey tool administered quarterly using the Qualtrics platform. Questions directly assess learning and support strengths and weaknesses, attitudes to research and beliefs over who should be involved in this research. Responses are linked, so changes to responses can be tracked over the duration of the study, allowing investigators to describe the effect of PAR on both researchers and participants with lived experience.

### Population and participation

The potential involvement of people with cognitive impairment carries additional ethical considerations.[25 26] Involvement of people living with dementia is actively encouraged and supported to maximise inclusivity, with levels of influence and support tailored to individual capabilities. People with a lived experience of dementia will be involved with all aspects of research design and implementation. The establishment and operation of the SRG will be recorded and analysed to demonstrate the changes in the wider research team and the influence of PAR on final research recommendations. Evidence of personal development over the project will be collected through the dynamic learning needs assessment, a quarterly survey completed by all members of the SRG, investigation team and project team. Evidence of impact will be collected through documentation of meetings and correspondence between consumer participants and the investigation team. Methods of involving people, reported personal development and other impacts and outcomes will be reported using STARDIT.[19]

### Stakeholder Reference Group

The SRG will guide participatory action throughout the AHR phases. The main role of the SRG is to ensure the measures including in the COM-IC recommendations are meaningful and relevant to people living with dementia while being practical and sustainable for care providers. The SRG will include a diverse range of stakeholders that reflects the heterogeneity of different groups affected by dementia.

### Eligibility

The SRG will have between eight and 15 members, including at least four people who have either had an experience of using dementia care and services in the last 10 years, or have been affected by dementia (including parents, loved ones and care partners). The SRG will also include at least one person who identifies as Aboriginal or Torres Strait Islander, at least two people with professional dementia research or health professional experience and at least two people with a professional experience of working in dementia care. SRG members will be expected to speak from their personal experience and, where appropriate, collate the views of people with an experience of dementia. SRG members will require a working knowledge of basic computer software—training will be provided as required to support participation.

### Recruitment and participation

Members of the SRG will be recruited through an online expression of interest, promoted through the professional and industry networks of the research team and established dementia support networks. Expressions of interest will be reviewed by a subset of the research team, including two consumer investigators. Members will be selected based on their experience, ability to represent a diverse population and the time they are able to commit

to SRG activities. Once informed consent is received from all SRG members, and any competing or conflicting interests have been declared and publicly reported, SRG activities will commence. Potential SRG members must have sufficient cognitive capacity at the time of their recruitment to provide informed consent—individuals incapable of providing consent will not be invited to join the SRG. Potential SRG members with a diagnosis of dementia or cognitive impairment will be contacted by a member of the investigation team with experience working with people living with dementia, who will discuss what SRG participation involves, address any concerns and confirm informed consent to participate. Acknowledging that dementia is a degenerative condition with variable rates of decline in cognitive capacity, consent will be confirmed and assented at each project activity.

### Working groups

Task-specific working groups may be established to seek broader input and experiences at some stages of the COM-IC project. An online expression of interest process similar to that for the SRG will be used to identify participants for each working group. Depending on the activity's objectives, eligibility criteria may include specific research experience, experience of living with dementia and/or experience of providing informal or formal care for a person living with dementia.

### Outcome measures
#### Primary outcome

The recommended COS is appropriate to the Australian aged care setting:

► Clarity—what and how outcomes are measured is understood by people with data collection and reporting responsibilities.
► Validity—outcome measures are sensitive to changes in determinants of quality care provision in routine care settings.
► Measurability—outcome measures are easily integrated into existing data collection routines and are elements that can be reliably quantified.
► Consistency—outcome measures are collected and reported the same way across different settings and organisational structures.
► Value—outcome measures are comprehensive and have a low administrative burden.

#### Secondary outcomes

Secondary outcomes will include:

► Effectiveness of the PAR methodology:
 SRG membership is sustained.
 SRG members contribute at each stage of the research programme.
 SRG is able to communicate ideas and decisions to the investigators.
 SRG endorses the final core outcome recommendations.

► Changes in stakeholder and researcher capacity, perspectives and beliefs measured using a dynamic learning needs analysis:
 Changes in attitudes.
 Changes in learning and support needs.
 Changes in beliefs.
 Changes in skills.

### Data collection

Data will be collected by members of the investigation team and industry partners. Data collected by the investigation team will include information about the PAR process, SRG and working group participants and activities. Data collected by industry partners will include information related to the feasibility study (outcome measures, implementation, acceptability). All data collection will be electronic.

### Data management

Data collected for the COM-IC project will be managed by the project team at The University of Queensland. Data will be protected by multifactor authentication across two platforms: Microsoft Teams, and the secure Research Data Manager (RDM) developed by The University of Queensland and cited as an approach to good research data management practice.[27] All data will be regularly backed up as part of the RDM. Password protection and multifactor authorisation will be used to ensure that only approved members of the research and project teams will be able to access data. Any identifiable data will be deidentified prior to any public dissemination. Any information collected by partner organisations will be securely stored under password and two-factor authorisation until it can be deidentified and transferred to The University of Queensland.

### Quality control

The secretariat will routinely evaluate data collection and storage and maintain the RDM record. Any adjustments required will be communicated with the group and regular data reports presented at quarterly meetings.

### Analysis plan

The final data analysis plan will be codeveloped with the SRG and is dependent on the outcome measures identified during recommendation development. Data analysis will be jointly conducted by the research team, SRG and working groups using agreed methods appropriate to data type. Stakeholders will be involved in data analysis and checking, developing conclusions, and reporting outcomes. Any conclusions from the analysis will be collectively agreed and disseminated in a publicly accessible way, including open access, peer-reviewed publications. Analysis will focus on determining the feasibility of implementing core outcome measures for dementia care, the impact and effectiveness of PAR and the data collected by the COS.

Feasibility analysis will be jointly conducted by the investigation team and a feasibility working group that includes

members of the SRG. The methodology includes tools such as surveys and interviews to measure change readiness, beliefs, attitudes and behaviours of system actors that influence implementation, measures of fidelity and time and cost of recommendation implementation.

Impact and effectiveness of PAR will be analysed using data on *level of engagement* (eg, meeting attendance, participation in activities, contributions and roles undertaken), *change in research capacity* (based on the dynamic learning needs assessment) and *ability to achieve consensus* (using tools such as a modified e-Delphi method). The overall PAR methodology will be reported using STARDIT.[19] SRG and working group impact on the COM-IC project will be evaluated using qualitative methods, including data mapping and familiarisation; transcription; coding; searching for themes; reviewing themes with study team members; labelling and summarising themes; and reporting the findings. In order to enhance validity of the analysis, two study team members will independently analyse the data thematically, with the analysis then checked for validity ('member checked') by a third study team member.[28–30] STARDIT-PM (Standardised Data on Initiatives - Preference Mapping tool) categories will be used to organise data into predefined 'super-categories' that allow consistent comparison with other initiatives reporting with STARDIT.[19]

COS data analysis methods will be codeveloped by the SRG and investigation team and will incorporate suitable quantitative methods, for example, descriptive statistics such as mean, SD, IQR and baseline comparisons.

## DISCUSSION

Improving care for people living with dementia requires timely and reliable diagnosis, empowerment and re-enablement of each individual in participating and planning their care, speedy translation of clinical evidence into practice and policy, and effective coordination of care provision across settings. Rapid translation of evidence into research remains a challenge due to the complexity of healthcare and social care services, limited funding resources for services, and lack of consensus on what constitutes 'good value care' between individuals living with dementia, service providers, and funders.

Rapid translation of COM-IC project findings is feasible as many outcome measures are already embedded in existing information systems and used as key performance indicators or accreditation standards by aged care providers,[5 31] and/or used in dementia research (trials, cohort or implementation studies of innovative care delivery models).[4 17 32–34]

The outcome of this project will be recommendations for a set of core outcome measures for routine care of people living with dementia that encompass both consumer preferences and international evidence-based care, and are tested and suitable for use in the aged care setting for people living with dementia.

There are some limitations that may impact the generalisability of our study and complicate translation of our research into practice. In the first instance, there are numerous sources of outcome measures that may be difficult to condense sufficiently and maintain the coverage required for the breadth and complexity of routine care provision. As there has not been consistency in outcome measurement for routine care, it is not possible to compare whether the outcome set developed is superior to any other. For the feasibility study, the time for data collection may not be adequate to demonstrate sensitivity of the outcome measures to changes in care provision, as some indicators take a longer period of time to show change. Finally, our SRG is designed to be representative of large segments of the community, but in itself comprised individuals that may not necessarily hold the beliefs and values of the entire community they represent.

## ETHICS AND DISSEMINATION
### Research ethics approval
This study has received ethical approval from The University of Queensland (HREC 2021/HE001932) and has received approval in kind from partner organisations. Any modifications to the protocol that may impact on the conduct of the study, including changes of study objectives, study design, study population, sample sizes, study procedures or significant administrative aspects, will be formalised as a protocol amendment, subject to HREC approval and subsequent notifications to authorities in accordance with local regulations.

### Additional considerations for potential cognitive impairment
Accommodations for people with varying levels of cognitive ability, including the potential for deterioration over the course of the study, have been discussed. In the '*Blueprint for dementia research*', WHO highlights the importance of involving people with lived experience at all stages of the research process, noting it is part of their rights under United Nations Convention on the Rights of Persons with Disabilities and other human rights instruments.[1] Involvement ensures the most efficient translation of research into practice, producing outcomes that are relevant to the people they most affect.

In line with the WHO blueprint, additional consideration has been given for accessibility, data sharing, capacity building, strengthening support networks, using existing advocacy and support structures, use of technology, and sharing knowledge.[1] Information is packaged in different ways and able to be tailored to the level of comprehension of the individual. Ongoing dynamic learning needs assessment monitors progress and motivation within the project team. Outcomes of the research will be published and distributed widely along familiar routes including academic and industry publications, symposia, and conference presentations. Audiences are diverse, including people with lived experience, clinicians, care professionals, aged care providers, advocacy groups, and

academics. Assent processes are used to confirm ongoing understanding of the research process. Representation of people with lived experience at every stage is factored into the design, as well as strategies for addressing any discomfort or potential harm as it is identified. Once the SRG is established, members will distribute findings through their networks and recruit working group members from the dementia community. Any findings will be distributed through SRG networks, meaning the broader dementia community has rapid access to results and has a familiar conduit to provide feedback in real time to the research team over the course of the project.

## Consent and assent

Prior to commencing any research activities, all people included in the study will receive a written participant information sheet and will sign a consent form. They will have the opportunity to ask any questions. For people with potential cognitive impairment, pictorial versions have been developed. As dementia is a progressive condition with uncertain prognosis, assent is requested prior to any recording of information. Participants are not required to participate against their will and are free to withdraw at any time.

## Confidentiality

All potentially sensitive information will be stored securely on The University of Queensland's research data management system (RDM). This system complies with all international data safety requirements. All project-related data will be stored in a durable format alongside project metadata, which will be regularly backed up by secure ITS (Internet Transaction Server). Data are accessible only to individuals listed on the project record and are accessible only via institutional usernames and passwords. Information published in reports, academic papers and presentations will be deidentified. There is a small risk of identification through familiarity explained to participants in the consent process.

## Dissemination of research findings

Results will be disseminated through networks established over the project, and in accordance with both the publication schedule and STARDIT reports.

## Data statement

Full access to outputs will be made available through UQ eSpace, an open access repository hosted by The University of Queensland.

## Author affiliations

[1]Centre for Health Services Research, The University of Queensland, Brisbane, Queensland, Australia
[2]Global Brain Health Institute, Trinity College Dublin, Dublin, Ireland
[3]WA Centre for Health and Ageing, The University of Western Australia, Perth, Western Australia, Australia
[4]Department of Health and Data Science, University of Liverpool, Liverpool, UK
[5]School of Public Health, University of Sydney, Sydney, New South Wales, Australia
[6]HammondCare International, London, UK
[7]HammondCare, Sydney, New South Wales, Australia
[8]Science for All, Melbourne, Victoria, Australia
[9]La Trobe University, Melbourne, Victoria, Australia
[10]Faculty of Health and Behavioural Sciences, The University of Queensland, Brisbane, Queensland, Australia
[11]Bolton Clarke Research Institute, Bolton Clarke, Brisbane, Queensland, Australia
[12]School of Public Health and Preventive Medicine, Monash University, Melbourne, Victoria, Australia
[13]Wynnum Manly Dementia Alliance, Brisbane, Queensland, Australia
[14]Bolton Clarke Research Institute, Brisbane, Queensland, Australia

**Contributors** Concept: developed through collaboration and communication between TC, KN, LG, LF, PW, SD, CC, TM, JN, DT, OPA, AW, JAL, JQ, GP and SEK. Design: all authors contributed to design ideas, finalised by TC, AW and DK. Component leads: overall project—TC; international review of core outcome measures led by AK; recommendation development led by TC and SEK; industry analysis led by TM and AM, supported by CC and JAL; gap analysis conducted by T-HD-T, assisted by DK, AK, TM and AM. Recommendation development: SEK and TC; feasibility study: JAL and CC, supported by AM and TM. Stakeholder Reference Group: JN, supported by DK. Industry contributions: LF and OPA represent hospital care; CC, TM, JAL and AM represent residential aged care and home care for older people; CC, SEK and LF represent advocacy and dementia support networks. DT, KN, AW, SD and PW gave expertise and insight on core outcome measures and their applications. Stakeholder Reference Group establishment and analysis; JN, DK, AW, TC. Writing: TC, DK, AW. Editing: all authors assisted with editing and proofreading the manuscript for final submission.

**Funding** This research is funded by a Medical Research Future Fund (Dementia, Ageing and Aged Care Mission) targeted grant, awarded in 2021 (2007650).

**Competing interests** JN is the volunteer director of the registered charity 'Science for All' and is paid as an individual by the charity for his work on this MRFF-funded project by written agreement. Science for All will receive funding through the COM-IC MRFF grant to conduct elements of the research programme. All funding is reported annually through the Australian Charities and Not-for-profits Commission (ACNC).

**Patient and public involvement** Patients and/or the public were involved in the design, or conduct, or reporting, or dissemination plans of this research. Refer to the Methods section for further details.

**Patient consent for publication** Not applicable.

**Provenance and peer review** Not commissioned; externally peer reviewed.

### ORCID iDs

Kim Nguyen http://orcid.org/0000-0002-2592-9372
Osvaldo P Almeida http://orcid.org/0000-0002-8689-6199
Danelle Kenny http://orcid.org/0000-0001-7396-9742
Judy A Lowthian http://orcid.org/0000-0002-9780-5256

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
