## [Reviewer comments · BMJ Open]

ARTICLE DETAILS

TITLE (PROVISIONAL)	Measuring the success of programs of care for people living with dementia – a protocol for consensus-building with consumers to develop a Core Outcome Set for Improving Care (COM-IC)
AUTHORS	Comans, Tracy; Nguyen, Kim; Gray, Len; Flicker, Leon; Williamson, Paula; Dodd, Susanna; Kearney, Anna; Cunningham, Colm; Morris, Thomas; Nunn, Jack; Trepel, Dominic; Almeida, Osvaldo; Kenny, Danelle; Welch, Alyssa; Lowthian, Judy; Quinn, John; Petrie, Glenys; Dao-Tran, Tiet-Hanh; Manchha, Asmita; Kurrle, Susan

VERSION 1 – REVIEW

REVIEWER	Handley, Melanie University of Hertfordshire, CRIPACC
REVIEW RETURNED	11-May-2023

GENERAL COMMENTS	Thank you for the opportunity to review the protocol for developing core outcome measures for improving care for people living with dementia. This is an important study that will develop a set of consistent and comparable measures that are feasible to use. The manuscript is well written and I have not further comments for the authors. I wish them well in their submission and the study.
--

REVIEWER	Zidarov, Diana CIUSSS Centre-Sud-de-l'Ile-de-Montreal
REVIEW RETURNED	18-May-2023

GENERAL COMMENTS	General comment: Thank you for the opportunity to review this exciting and relevant protocol aiming to improve the quality of dementia care using a participatory action approach embedded in the Alignment, harmonization, and results framework. I understand this protocol is a general overview (research plan) of a project comprising multiple studies aiming to develop a core outcome set for and with different stakeholders, including people living with dementia. However, much methodological information is missing that does not allow the reader to determine whether this goal will be achieved by using appropriate methods for each study included in the project (e.g., identifying a set of proposed outcome measures; developing guidelines for core outcome measures in dementia care; feasibility study). My main suggestion is to add specific methods with more details for each study integrated into the AHR Framework. Specific comments Introduction: the importance of developing COS is very well explained, in addition to including the voice of people living with
--

	dementia; however, it is not mentioned if a COS has already been developed in dementia care. This information is needed to justify the need to develop or adapt one. Line 32-33, P6: it would have been relevant to know if a core outcome set for dementia already exists; if not, what methods will be used to develop one and if yes, why not use the existing ones? Although relevant to consult industrial partners about the outcome measures currently used in the Australian context, please justify why only this approach is chosen and why a search of existing recent systematic reviews of outcome measures used in dementia care will not be performed, or, if not available, the conduct of such a systematic review will not been done. In the proposed approach, some relevant outcome measures may be missed. Please justify why you are not using recommendations/guidance on developing core outcome sets, such as the COMET initiative listed in the reference list. Table 1: it appears that the word indicator is used interchangeably with outcome measure; please specify if these concepts are the same, and if not, please provide a definition. Line 37-38, P7: it is stated that a standardized set of guidelines for core outcome measures will be generated in the harmonization step. How will the guidelines be developed? What are precisely the goals or objectives of these guidelines? Who are the intended end users? What scientific literature will be consulted? An explicit method of developing these guidelines should be provided. Another issue in this section is that the authors mention that the developed guidelines will guide the selection of core outcome measures for specific populations/care settings/interventions with dementia. Does that mean that more than one core outcome set would be developed? This aspect needs further justification, information, and discussion. This section also states that the guidelines will be accompanied by an implementation plan that will address the barriers to implementing a core outcome set of measures. It is not mentioned how these barriers will be assessed (methods) and how the strategies addressing the barriers will be identified and developed; in addition, will these strategies be tailored according to the different care delivery models and care settings that provide services for this heterogenous population? More information is also required about the feasibility study (e.g., study design, recruitment, feasibility outcomes to be measured, etc.). Also, what are exactly the goals of the feasibility study: uptake of the core outcome set or some other objectives, the practicality and sustainability of the guidelines and implementation plan are also mentioned. Information needs to be provided, and how will this be assessed? P8, line 3: how many organizations are targeted in the feasibility study? Outcome measures section: Please revise this section. The primary outcome looks like a deliverable, i.e., the creation of a guideline; the secondary outcomes are more study objectives. Please provide references and more information on the development and content of the dynamic Learning Needs Analysis, given that it would be used to measure the change in capacity, perspectives, and beliefs.
--	--

REVIEWER	Vishnu, Venugopalan All India Institute of Medical Sciences, Neurology
REVIEW RETURNED	25-Jul-2023

GENERAL COMMENTS	The authors describe the protocol of their COM-IC project which aims to identify suitable core outcome measures. They look at acceptability and feasibility from the point of view of stake holders like people living with dementia, formal and informal carers, age care industry, researchers, clinicians and policy makers. Few points for authors consideration:  1. Authors should discuss the limitations of the study. At present its a single line under strengths and limitations 2. In the SRG - it's mentioned that "Aboriginal or Torres Strait Islander - with or without dementia". It will be better to have at least one person with dementia. 3. The feasibility study methodology may be described in detail.
---

VERSION 1 – AUTHOR RESPONSE

Response to Reviewers

Reviewer 1

Thank you for the opportunity to review the protocol for developing core outcome measures for improving care for people living with dementia. This is an important study that will develop a set of consistent and comparable measures that are feasible to use. The manuscript is well written and I have not further comments for the authors. I wish them well in their submission and the study.

We thank the author for their time in reviewing our paper and their support for this publication.

Reviewer 2

General comment: Thank you for the opportunity to review this exciting and relevant protocol aiming to improve the quality of dementia care using a participatory action approach embedded in the Alignment, harmonization, and results framework. I understand this protocol is a general overview (research plan) of a project comprising multiple studies aiming to develop a core outcome set for and with different stakeholders, including people living with dementia. However, much methodological information is missing that does not allow the reader to determine whether this goal will be achieved by using appropriate methods for each study included in the project (e.g. identifying a set of proposed outcome measures; developing guidelines for core outcome measures in dementia care; feasibility study). My main suggestion is to add specific methods with more details for each study integrated into the AHR Framework.

Thank you for your time in reviewing our protocol. It is heartening to see that readers can clearly understand the objectives of the study. We understand and agree that the methodology is a sketch at this point, but that is due to the nature of co-design, where our stakeholders and investigators make design decisions considered from multiple perspectives. Consequently, the level of methodological detail usually seen in protocols has not yet been developed. **We have modified the methodology section to make this clearer from the outset and to elaborate further on co-design and participatory action research.**

Introduction: the importance of developing COS is very well explained, in addition to including the voice of people living with dementia; however, it is not mentioned if a COS has already been developed in dementia care. This information is needed to justify the need to develop or adapt one.

Thank you for highlighting the substantial amount of work that has already been done in developing core outcome sets, we acknowledge the complicated balance between comprehensive assessment in

the provision of quality care and the standardisation required for effective international comparison. At this early protocol stage, the full extent of existing COS has not been scoped or documented (this will occur during the Alignment phase) and we are uncertain if there are any COS for dementia that are specifically relevant to routine care settings in Australia. The scoping review in the Alignment phase will produce the scaffolding to introduce core outcome measurement to the stakeholders, who will likely have little former knowledge of this topic. **We have modified the introduction and methods to make this research question clearer and draw attention to our intention to answer it through the scoping review.**

Line 32-33, P6: it would have been relevant to know if a core outcome set for dementia already exists; if not, what methods will be used to develop one and if yes, why not use the existing ones? Although relevant to consult industrial partners about the outcome measures currently used in the Australian context, please justify why only this approach is chosen and why a search of existing recent systematic reviews of outcome measures used in dementia care will not be performed, or, if not available, the conduct of such a systematic review will not be done. In the proposed approach, some relevant outcome measures may be missed. Please justify why you are not using recommendations/guidance on developing core outcome sets, such as the COMET initiative listed in the reference list.

Thank you again for highlighting your concerns about adapting existing sets. We have clarified this information to demonstrate a full scoping review is being undertaken as part of the Alignment phase, and findings from this scoping review will inform selection of measures, in conjunction with information from industry. **We have also reworked the justification to illustrate why a triangulation approach and gap analysis is considered the most comprehensive way forward to examine and define core outcome measures for routine care in the Australian context.**

Table 1: it appears that the word indicator is used interchangeably with outcome measure; please specify if these concepts are the same, and if not, please provide a definition.

We appreciate your attention to detail here and confirm that these two concepts are different, and in this context here, we should have consistently used the term 'outcome measure.' **We have updated the table accordingly.**

Line 37-38, P7: it is stated that a standardized set of guidelines for core outcome measures will be generated in the harmonization step. How will the guidelines be developed? What are precisely the goals or objectives of these guidelines? Who are the intended end users? What scientific literature will be consulted? An explicit method of developing these guidelines should be provided. Another issue in this section is that the authors mention that the developed guidelines will guide the selection of core outcome measures for specific populations/care settings/interventions with dementia. Does that mean that more than one core outcome set would be developed? This aspect needs further justification, information, and discussion.

We interpret this comment to be an extension of the general comment and reiterate that we are consulting and co-designing all methods with the stakeholder reference group. The guidelines will be a set of recommendations for industry on the collection, reporting and interpretation of the outcome measures that are essential to evaluating the quality of care provided to people living with dementia. Although their development will be guided by the NHMRC handbook for developing guidelines, they will not be clinical guidelines per se. **We have amended the text to replace the term Guidelines with Recommendations to clarify this intent and have expanded the information provided on how these recommendations will be developed.**

This section also states that the guidelines will be accompanied by an implementation plan that will address the barriers to implementing a core outcome set of measures. It is not mentioned how these barriers will be assessed (methods) and how the strategies addressing the barriers will be identified and developed; in addition, will these strategies be tailored according to the different care delivery models and care settings that provide services for this heterogenous population? More information is

also required about the feasibility study (e.g. study design, recruitment, feasibility outcomes to be measured, etc.). Also, what are exactly the goals of the feasibility study: uptake of the core outcome set or some other objectives, the practicality and sustainability of the guidelines and implementation plan are also mentioned. Information needs to be provided, and how will this be assessed?

Identification of the barriers to implementation of a COS in Australian care settings will occur during the harmonisation phase. Once the COS has been agreed, engagement with our stakeholder reference group and industry partners will be conducted to identify barriers and co-design the methodology and criteria that will be used to evaluate the feasibility study in the Results phase. At this early protocol stage, we must adopt a flexible approach to evaluation methods, which will evolve through the co-design process between stakeholders, researchers and industry partners.

P8, line 3: how many organizations are targeted in the feasibility study?

We have two large industry partners with very different operational structures. **This has been clarified and described in the Results section** (Representatives from two large, Australian aged-care providers are embedded in the COM-IC investigator team).

Outcome measures section: Please revise this section. The primary outcome looks like a deliverable, i.e. the creation of a guideline; the secondary outcomes are more study objectives. Please provide references and more information on the development and content of the dynamic Learning Needs Analysis, given that it would be used to measure the change in capacity, perspectives, and beliefs.

We have revised the outcome measures section and expanded on the background of dynamic learning needs assessment.

Reviewer 3

1. Authors should discuss the limitations of the study. At present its a single line under strengths and limitations

Thank you for your interest in the strengths and limitations of our study. **We have expanded our discussion to include key limitations, such as the breadth and complexity of information to be condensed, the short duration of the feasibility study, and the potential for our stakeholder reference group to be insufficiently representative.**

2. In the SRG - it's mentioned that "Aboriginal or Torres Strait Islander - with or without dementia". It will be better to have at least one person with dementia.

Thank you for your attention to diversity. Aboriginal and Torres Strait Islander peoples represent 3% of the Australian population, but experience many health conditions differently for a variety of different reasons. While it is certainly desirable for us to include a person of Aboriginal and Torres Strait Islander descent who lives with dementia and is both willing and able to engage with our study for the two year period, we are doubtful a single individual will be able to fulfil all of these criteria. Even without knowledge of dementia, an Indigenous Australian perspective would be a valuable addition to our research. The tag 'with or without dementia' was included for clarity. Given the reader's interpretation, **we have removed the sub-clause.**

3. The feasibility study methodology may be described in detail.

Thank you for your suggestion. We have discussed this apparent gap at length but note that it is not possible or desirable to predict the outcomes of the participatory action co-design process. Our stakeholder reference group was not formed prior to the publication of the protocol, so we are unable to predict the precise methodology that will allow the stakeholders and investigators to answer the research question. This protocol has been written to highlight our co-design methodology rather than detailing possible components of the three phases. **As such, we have modified the introduction and methodology to focus more clearly on participatory action and co-design.**